# The G allele of the *IGF1* rs2162679 SNP is a potential protective factor for any myopia: Updated systematic review and meta-analysis

**Bo Meng** *, **Kang Wang, Yingxiang Huang** , **Yanling Wang**

Beijing Friendship Hospital, Capital Medical University, Beijing, China

* mengbo-2007@163.com

## Abstract

### Background

The insulin-like growth factor 1 (*IGF1*) gene is located within the myopia-associated MYP3 interval, which suggests it may play an important role in the progression of myopia. However, the association between *IGF1* SNPs and any myopia is rarely reported.

### Methods

A comprehensive literature search was conducted on studies published up to July 22, 2021 in PubMed, EMBASE, CBM, COCHRANE, CNKI, WANFANG and VIP databases. Odds ratios (ORs) and 95% confidence intervals (CIs) were calculated for single-nucleotide polymorphisms (SNPs) that have been evaluated in at least three studies.

### Results

Nine studies involving 4596 subjects with any myopia and 4950 controls examined 25 SNPs in *IGF1* gene, among which seven SNPs were included in this meta-analysis. Significant associations were not found in any genetic models between rs6214, rs12423791, rs5742632, rs10860862, rs5742629 and any myopia. Rs2162679 was suggestively associated with any myopia in the codominant model (GA vs. AA: OR = 0.87, 95% CI: 0.76–1.00) and the dominant model (GG+GA vs. AA: OR = 0.88, 95% CI = 0.78–1.00).

### Conclusion

Meta-analysis of updated data reveals that the G allele of the *IGF1* rs2162679 SNP is a potential protective factor for any myopia, which is worth further researches.

## Introduction

Recently, myopia has emerged as a major public health concern worldwide. In the last several decades, the prevalence of myopia in the United States and Europe has increased [1, 2]. Asian countries have the highest rates of myopia, especially in east and Southeast Asia [3]. In China,

**Data Availability Statement:** All relevant data are within the paper and its Supporting Information files.

**Funding:** The authors received no specific funding for this work.

**Competing interests:** The authors have declared that no competing interests exist.

Singapore and Taiwan, the prevalence of myopic subjects aged 12–39 years has rapidly increased to 67–96% [4–6]. Because of its higher prevalence, myopia imposes enormous economic and social burdens worldwide [7].

Although myopia is classified as a benign disorder that can be corrected with optical modalities, myopic eyes with a long axial lengths ($\geq$26 mm) or a high degree of myopic refractive error ($\leq$−6D), can cause blindness with complications such as glaucoma, macular degeneration, retinal detachment, myopic foveoschisis, and choroidal neovascularization [8, 9]. Myopia has already become the second most common cause of legal blindness [10, 11]. Therefore, it is very important to identify the potential risk factors to establish preventive strategies for myopia.

The pathogenesis of myopia remains unclear. Research has shown that myopia is a multifactorial disease that results from an interaction between environmental and genetic factors [12–14]. Environmental factors include near work, outdoor activities, level of education, light exposure, diet and urbanization [15, 16]. For example, in two independent population-based cohorts of individuals from European descent, Verhoeven et al. [17] found that the genetic risk of an individual for myopia is significantly affected by his or her educational level. Higher education affects myopia by increasing the amount of time spent doing near work activities [18]. By contrast, children who spend more time engaged in outdoor activities have shown a reduced prevalence and a slower progression of myopia. Although the environment plays a role in the progression of myopia, results of twins and family-based studies have shown that the genetic component is significant [19, 20]. Association studies have led to the identification of many susceptibility and causative genes for myopia. These genes are enriched for certain functional annotations, such as neurotransmitter functions (GRIA4), ion channel activity (KCNQ5, CD55 and CACNA1D), retinoic acid metabolism (RDH5, CYP26A1 and RORB), extracellular matrix remodeling (LAMA2 and BMP2) and ocular development (SIX4, CHD7 and PRSS56) [21].

The *IGF1* gene is located in 12q23.2 of the human genome and contains six exons [22]. One of the proteins encoded by this gene is similar to insulin in its structure and function. Previous animal studies showed that the *IGF1* gene contributed to eye development and disease. For example, *IGF1/FGF2*-treated eyes in animal studies could have an increased vitreous chamber depth, decreased anterior chamber depth, and changes in the sclera [23]. Hellstrom et al. showed that lack of *IGF1* in knockout mice prevented normal retinal vascular growth by preventing VEGF-induced activation of protein kinase B, a kinase that is critical for endothelial cell survival [24]. Additionally, Ruberte et al. [25] suggested that *IGF1* played a role in the development of ocular complications in patients with diabetes for a long period of time. The *IGF1* gene also is located within the myopia-associated MYP3 interval, which has been mapped using the linkage disequilibrium method. This suggests that *IGF1* may play an important role in the progression of myopia. However, the association between *IGF1* SNPs and any myopia is rarely reported. Therefore, we present herein an updated systematic review and meta-analysis to evaluate the potential association between *IGF1* SNPs and any myopia.

## Methods

### Search strategy

The review protocol was registered with the International Prospective Register of Systematic Reviews (PROSPERO, CRD42021274322) and performed according to the Preferred Reporting Items for Systematic review and Meta-Analyse Statement (PRISMA) guidelines. We searched the following databases: PubMed, EMBASE, Cochrane Library and several Chinese databases, such as the Chinese biomedical literature database (CBM), China National

Knowledge Infrastructure (CNKI), WANFANG DATA and VIP database from their inception to July 22, 2021. The selected key words were used as free words, truncations and MeSH terms. Reference lists from the retrieved articles were manually screened for potential articles, if any, that had not been captured by the electronic search. No language restrictions were applied throughout the search process.

### Inclusion and exclusion criteria

Inclusion criteria were as follows: 1) original case-control or family-based studies that evaluated the association between polymorphisms of *IGF1* and any myopia; 2) numbers or frequencies in case and control groups reported for each genotype or allele; 3) if the study was reported in duplicate, the version with the most comprehensive content was included; and 4) studies including normal individuals with spherical equivalent refraction that ranged from -1.5 to 1.5 diopters and were free from any complications.

Exclusion criteria were as follows: 1) animal studies, reviews, conference proceedings, case reports, editorials; and 2) articles providing incomplete data or that could not be acquired through various means.

### Data extraction

Two independent authors screened all retrieved records and made decisions on which studies to include. Any disagreements were resolved by discussion. Further, any uncertainties were resolved by consultation with a third author. The information of first author, year of publication, ethnicity, genotyping type, sample size, polymorphisms studied, genotype distribution, minor allele, Hardy–Weinberg equilibrium (HWE) and conclusions on any myopia association were collected. If allele data were not available in original reports, they were calculated based on genotypic data.

### Assessment of study quality

Study quality was assessed using revised criteria according to Little's recommendations [26] for gene-disease associations, with an aim to investigate potential bias in summary results. These criteria included: 1) the genotyping method used; 2) definition of cases and methods of ascertainment; 3) socio-demographic characteristics of subjects; 4) confounding factors mentioned in articles; and 5) confidence intervals of genotype frequency. An overall quality score was generated, and studies with a score $\geq 3$ were considered to have high quality.

### Statistical analysis

All statistical analyses were performed using RevMan 5.3. Association of each SNP with myopia in pooled samples, along with pooled odds ratios (ORs) and 95% confidence intervals (95% CIs), were evaluated. The $I^2$ statistic was used to quantify heterogeneity. In addition, funnel plot was used to evaluate the publication bias.

## Results

### Eligible studies and study characteristics

A total of 145 potentially relevant articles were retrieved. Ultimately, nine studies that met all criteria were included for this meta-analysis (Fig 1) [27–35].

Overall, 25 SNPs associated with the *IGF1* gene were investigated at least once in nine studies. Among these SNPs, seven were tested in at least three studies and then were included in the meta-analysis. The study subjects were Chinese [29, 31, 32, 34, 35], Japanese [27, 28],

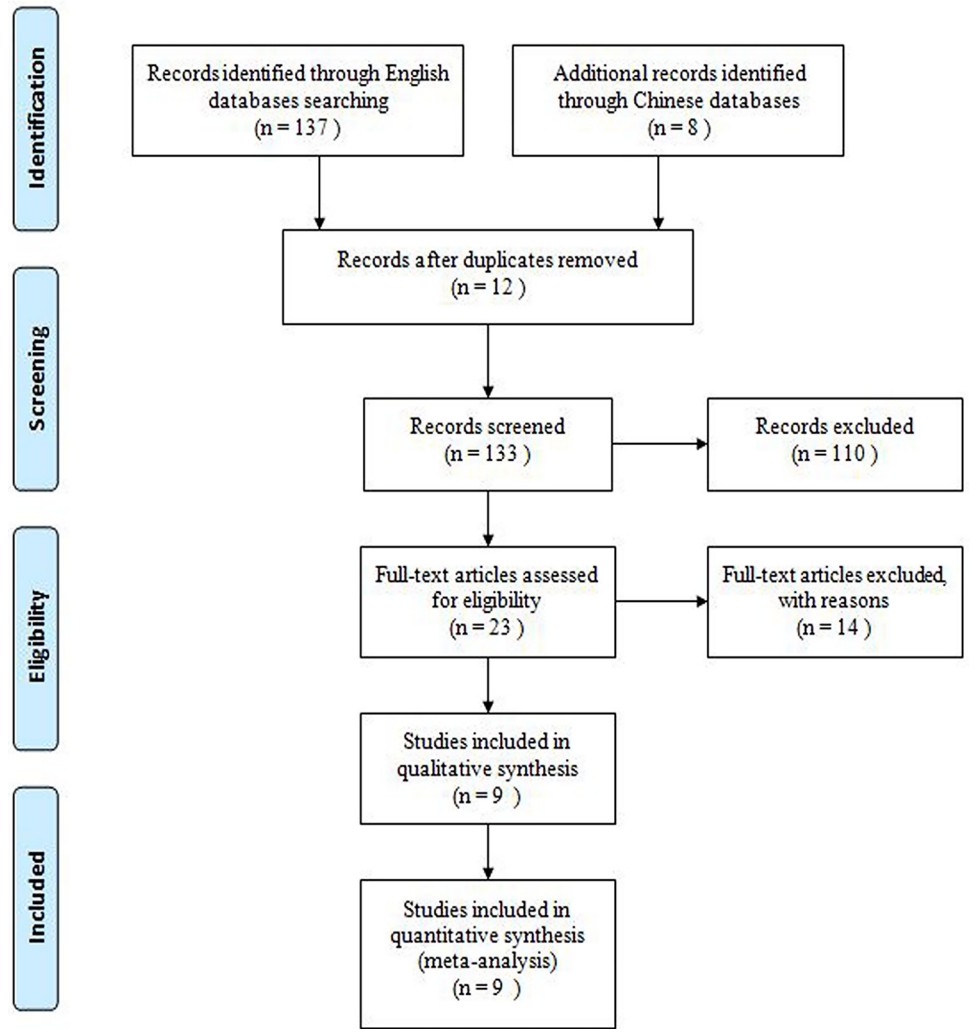

**Fig 1. Flowchart of study inclusion.**

Egyptian [33] and Polish [30] with sample sizes that ranged from 127 to 1339. The total sample size was 9546 (4596 individuals with any myopia and 4950 controls).

The methods of gene analysis included restriction fragment length polymorphism (RFLP), matrix-assisted laser desorption/ionization time-of-flight mass spectrometry (MALDI-TOF), RT-PCR, SnaPshot and polymerase chain reactionand ligase detection reaction (PCR-LDR). The quality scores of the included studies were greater than four, which indicated a favorable methodological quality. Table 1 summarizes the characteristics of the included studies.

## Association of IGF1 SNPs with any myopia

Rs2162679 was tested in three studies [27, 28, 32] with 2014 cases and 2048 controls. Fixed-effects models were used to calculate the pooled ORs. Our findings suggested that there were no significant associations for the allelic model (G vs. A: OR = 0.93, 95% CI: 0.85–1.02, $P = 0.14$), dominant model (GG+GA vs. AA: OR = 0.88, 95% CI = 0.78–1.00, $P = 0.05$), recessive model (GG vs. GA+AA: OR = 0.99, 95% CI = 0.82–1.19, $P = 0.92$ and codominant model

Table 1. Characteristics of all studies included in the meta-analysis.

| First author | Year | Ethnicity | Genotyping type | Quality score | SNP ID | Sample Case | Sample Control | Mean age(y) Case | Mean age(y) Control | Mean refractive errors (D) Case | Mean refractive errors (D) Control | Genotype Case 1/1 | Case 1/2 | Case 2/2 | Genotype Control 1/1 | Control 1/2 | Control 2/2 | Minor allele | HWE |
|---|---|---|---|---|---|---|---|---|---|---|---|---|---|---|---|---|---|---|---|
| **Cheng** | 2020 | Chinese | PCR-LDR | 5 | rs6214 | 281 | 373 | 9.84 ±1.55 | 8.06 ±1.43 | -2.55±1.64▲ | 0.84 ±0.81▲ | 59 | 140 | 82 | 89 | 186 | 98 | A | yes |
| | | | | | rs5742653 | | | | | -2.55±1.84△ | 0.88 ±0.83△ | 62 | 140 | 79 | 83 | 186 | 104 | G | yes |
| | | | | | rs4764697 | | | | | | | 9 | 83 | 189 | 10 | 102 | 261 | T | yes |
| | | | | | rs12423791 | | | | | | | 16 | 103 | 162 | 27 | 146 | 200 | C | yes |
| | | | | | rs2162679 | | | | | | | 29 | 122 | 130 | 52 | 175 | 146 | G | yes |
| | | | | | rs5742612 | | | | | | | 21 | 112 | 148 | 34 | 157 | 182 | C | yes |
| **Zidan** | 2016 | Egyptian | RFLP | 4 | rs5742632 | 136 | 272 | 41.2±9.0 | 42.23 ±8.0 | -4.41±1.42▲★,- 9.34±3.1▲ | 0.57 ±0.32▲ | 27 | 97 | 148 | 11 | 45 | 80 | C | N/A |
| | | | | | rs6214 | | | 40.7±8.7 | | -4.39±1.4△★,- 9.28±2.9△ | 0.59 ±0.31△ | 44 | 123 | 105 | 12 | 46 | 78 | A | N/A |
| **Wang** | 2016 | Chinese | SNaPshot | 5 | rs10860860 | 1244 | 1380 | 41.26 ±13.51 | 58.39 ±12.77 | -10.12±3.45▲ | N/A | 31 | 331 | 882 | 36 | 373 | 971 | T | no |
| | | | | | rs10860862 | | | | | -10.03±3.16△ | | 38 | 357 | 849 | 41 | 393 | 946 | T | no |
| | | | | | rs2946834 | | | | | | | 221 | 606 | 417 | 252 | 675 | 453 | T | yes |
| | | | | | rs6214 | | | | | | | 280 | 620 | 344 | 321 | 689 | 370 | A | yes |
| | | | | | rs12821878 | | | | | | | 3 | 121 | 1120 | 5 | 163 | 1212 | A | yes |
| | | | | | rs35766 | | | | | | | 130 | 525 | 586 | 186 | 596 | 598 | G | yes |
| **Zhao** | 2013 | Chinese | TOFMS | 5 | rs10860861 | 302 | 401 | 1.24 ±16.34 | 43.32 ±22.15 | -16.54±5.26▲ | 0.39 ±0.82▲ | 44 | 148 | 110 | 66 | 197 | 138 | C | yes |
| | | | | | rs10860862 | | | | | -16.39±5.47△ | 0.42 ±0.80△ | 8 | 84 | 210 | 12 | 117 | 272 | T | yes |
| | | | | | rs6214 | | | | | | | 89 | 145 | 68 | 101 | 200 | 100 | G | yes |
| | | | | | rs5742629 | | | | | | | 48 | 167 | 87 | 58 | 186 | 157 | G | no |
| | | | | | rs12423791 | | | | | | | 26 | 127 | 149 | 24 | 136 | 241 | C | yes |
| | | | | | rs35766 | | | | | | | 134 | 131 | 37 | 207 | 157 | 37 | G | yes |
| | | | | | rs1457601 | | | | | | | 18 | 130 | 154 | 21 | 140 | 240 | A | yes |
| **Miyake** | 2013 | Japanese | TaqMan | 4 | rs6214 | 1339 | 1194 | 57.2 ±14.9 | 50.3 ±15.9 | -12.69±4.54▲ | N/A | 277 | 641 | 373 | 268 | 585 | 341 | C | yes |
| | | | | | rs978458 | | | | | | | 256 | 661 | 361 | 264 | 596 | 334 | T | yes |
| | | | | | rs5742632 | | | | | | | 209 | 657 | 410 | 229 | 586 | 379 | G | yes |
| | | | | | rs12423791 | | | | | | | 97 | 452 | 672 | 85 | 468 | 641 | C | yes |
| | | | | | rs2162679 | | | | | | | 178 | 540 | 569 | 149 | 541 | 504 | C | yes |
| **Yoshida** | 2013 | Japanese | TaqMan | 5 | rs6214 | 446 | 481 | 37.9 ±11.9 | 39.3 ±11.0 | -11.7±2.24▲ | -1.5~ +1.5 | 58 | 205 | 183 | 55 | 215 | 211 | G | yes |
| | | | | | rs1111262 | | | | | -11.7±2.27△ | | 17 | 138 | 291 | 18 | 150 | 313 | A | yes |
| | | | | | rs972936 | | | | | | | 93 | 221 | 132 | 118 | 240 | 123 | G | yes |
| | | | | | rs5742629 | | | | | | | 70 | 214 | 162 | 94 | 237 | 150 | G | yes |
| | | | | | rs12423791 | | | | | | | 32 | 174 | 240 | 45 | 204 | 232 | C | yes |

*(Continued)*

Table 1. (Continued)

| First author | Year | Ethnicity | Genotyping type | Quality score | SNP ID | Sample | | Mean age(y) | | Mean refractive errors (D) | | Genotype distribution | | | | | | Minor allele | HWE |
|---|---|---|---|---|---|---|---|---|---|---|---|---|---|---|---|---|---|---|---|
| | | | | | | Case | Control | Case | Control | Case | Control | Case | | | Control | | | | |
| | | | | | | | | | | | | 1/1 | 1/2 | 2/2 | 1/1 | 1/2 | 2/2 | | |
| | | | | | rs2162679 | | | | | | | 44 | 193 | 209 | 55 | 215 | 211 | G | yes |
| | | | | | rs5742612 | | | | | | | 41 | 188 | 217 | 50 | 211 | 220 | C | yes |
| Zhuang | 2012 | Chinese | MALDI-TOF | 5 | rs10860861 | 421 | 401 | 38.29 ±16.57 | 68.77 ±10.65 | -14.57±5.6▲ | 0.39 ±0.82▲ | 153 | 202 | 66 | 138 | 197 | 66 | C | yes |
| | | | | | rs10860862 | | | | | -14.51±5.64△ | 0.42 ±0.8△ | 294 | 117 | 10 | 272 | 117 | 12 | T | yes |
| | | | | | rs6214 | | | | | | | 99 | 205 | 117 | 100 | 200 | 101 | G | yes |
| | | | | | rs5742629 | | | | | | | 128 | 222 | 71 | 157 | 186 | 58 | G | yes |
| | | | | | rs12423791 | | | | | | | 219 | 170 | 32 | 241 | 136 | 24 | C | yes |
| | | | | | rs35766 | | | | | | | 44 | 187 | 190 | 37 | 157 | 207 | G | yes |
| | | | | | rs1457601 | | | | | | | 217 | 180 | 24 | 240 | 140 | 21 | A | yes |
| Mak | 2012 | Chinese | RFLP | 5 | rs12579077 | 300 | 300 | 18−45 | 18−45 | ≤−8.0 | −1.0~ +1.0 | 38 | 109 | 153 | 36 | 128 | 136 | C | yes |
| | | | | | rs35767 | | | | | | | 46 | 126 | 128 | 47 | 134 | 119 | T | yes |
| | | | | | rs4764698 | | | | | | | 30 | 115 | 155 | 28 | 128 | 144 | C | yes |
| | | | | | rs12423791 | | | | | | | 29 | 132 | 139 | 30 | 135 | 135 | G | yes |
| | | | | | rs7956547 | | | | | | | 5 | 83 | 212 | 5 | 74 | 221 | G | yes |
| | | | | | rs5742632 | | | | | | | 62 | 150 | 88 | 58 | 153 | 89 | C | yes |
| | | | | | rs2373721 | | | | | | | 6 | 80 | 203 | 7 | 80 | 213 | G | yes |
| | | | | | rs6539035 | | | | | | | 5 | 78 | 217 | 6 | 71 | 223 | G | yes |
| | | | | | rs6214 | | | | | | | 74 | 146 | 80 | 85 | 137 | 78 | A | yes |
| | | | | | rs5742723 | | | | | | | 30 | 118 | 152 | 31 | 127 | 142 | A | yes |
| Rydzanicz | 2011 | Polish | RFLP | 4 | rs6214 | 127 | 148 | 27.1 ±22.63 | 38.6 ±18.54 | -2.75±2.00▲ | -0.03 ±1.26 | 22 | 72 | 64 | 16 | 78 | 54 | A | yes |
| | | | | | rs10860860 | | | 40.2 ±20.43 | | -9.32±3.89△ | | 18 | 68 | 72 | 13 | 75 | 60 | T | yes |
| | | | | | rs2946834 | | | | | | | 19 | 62 | 75 | 14 | 61 | 72 | T | yes |

HWE: Hardy-Weinberg Equilibrium; N/A: Not available; ▲Right eye

△Left eye

★Simple myopia

High-grade myopia; 1/1: genotype with homozygous allele 1; 1/2: genotype with heterozygous alleles; 2/2: genotype with homozygous allele 2.

(GG vs. AA: OR = 0.92, 95% CI = 0.76–1.13, $P$ = 0.43). There were suggestive associations for the codominant model (GA vs. AA: OR = 0.87, 95% CI = 0.76–1.00, $P$ = 0.04) (Fig 2, Table 2).

Rs6214 was tested in nine studies [27–29, 31–36] with 4715 cases and 4814 controls. Random -effects models were used to calculate the pooled ORs. Our findings suggested that there were no significant associations for the allelic model (A vs. G: OR = 0.98, 95% CI: 0.91–1.06, $P$ = 0.64), dominant model (AA+AG vs. GG: OR = 1.03, 95% CI = 0.90–1.18, $P$ = 0.65), recessive model (AA vs. AG+GG: OR = 1.00, 95% CI = 0.89–1.11, $P$ = 0.94 and codominant model (AA vs. GG: OR = 1.02, 95% CI = 0.87–1.20, $P$ = 0.82 and AG vs. GG: OR = 1.02, 95% CI = 0.90–1.15, $P$ = 0.73) (Fig a in S1 File, Table 2).

Rs12423791 was tested in six studies [27–29, 31, 32, 35] with 2971 cases and 3150 controls. Random-effects models were used to calculate the pooled ORs. Our findings demonstrated that there were no significant associations between rs12423791 and any myopia in the allelic model (C vs. G: OR = 0.95, 95% CI: 0.81–1.11, $P$ = 0.51), dominant model (CC+CG vs. GG: OR = 0.96, 95% CI = 0.80–1.16, $P$ = 0.68), recessive model (CC vs. CG+GG: OR = 0.92, 95% CI = 0.73–1.15, $P$ = 0.45 and codominant model (CC vs. GG: OR = 0.93, 95% CI = 0.71–1.22, $P$ = 0.61 and CG vs. GG: OR = 0.97, 95% CI = 0.82–1.16, $P$ = 0.76) (Fig b in S1 File, Table 2).

Rs5742632 was tested in three studies [28, 29, 33] with 1848 cases and 1630 controls. Fixed -effects models were used to calculate the pooled ORs. Our findings suggested that there were no significant associations for the allelic model (C vs. G: OR = 0.97, 95% CI = 0.88–1.07, $P$ = 0.57), dominant model (CC+CG vs. GG: OR = 1.01, 95% CI = 0.88–1.17, $P$ = 0.88), recessive model (CC vs. CG+GG: OR = 0.89, 95% CI = 0.75–1.07, $P$ = 0.22 and codominant model (CC vs. GG: OR = 0.91, 95% CI = 0.75–1.12, $P$ = 0.38 and CG vs. GG: OR = 1.04, 95% CI = 0.90–1.21, $P$ = 0.59) (Fig c in S1 File, Table 2).

Rs10860862 was tested in three studies [31, 34, 35] with 1967 cases and 2182 controls. Fixed-effects models were used to calculate the pooled ORs. Our findings demonstrated that there were no significant associations between rs10860862 and any myopia in the allelic model (T vs. G: OR = 1.02, 95% CI = 0.91–1.14, $P$ = 0.80), dominant model (TT+TG vs. GG: OR = 1.00, 95% CI = 0.87–1.16, $P$ = 0.98), recessivemodel (TT vs. TG+GG: OR = 1.06, 95% CI = 0.84–1.35, $P$ = 0.62 and codominant model (TT vs. GG: OR = 1.05, 95% CI = 0.73–1.51, $P$ = 0.81 and TG vs. GG: OR = 1.00, 95% CI = 0.86–1.16, $P$ = 1.00) (Fig d in S1 File, Table 2).

Rs35766 was tested in three studies [31, 34, 35] with 1964 cases and 2182 controls. Random-effects models were used to calculate the pooled ORs. Our findings suggested that there were no significant associations for the allelic model (G vs. A: OR = 0.93, 95% CI: 0.74–1.16, $P$ = 0.51), dominant model (GG+GA vs. AA: OR = 0.95, 95% CI = 0.69–1.31, $P$ = 0.77), recessive model (GG vs. GA+AA: OR = 0.81, 95% CI = 0.65–1.00, $P$ = 0.05 and codominant model (GG vs. AA: OR = 0.83, 95% CI = 0.56–1.21, $P$ = 0.32 and GA vs. AA: OR = 1.01, 95% CI = 0.77–1.32, $P$ = 0.97) (Fig e in S1 File, Table 2).

SNP rs5742629 was investigated in three studies [27, 31, 35] with 1169 cases and 1283 controls. Our findings indicated that no significant associations were present between this SNP and any myopia using the allelic model (G vs. A: OR = 0.94, 95% CI: 0.71–1.25, $P$ = 0.67), dominant model (GG+GA vs. AA: OR = 1.02, 95% CI = 0.65–1.59, $P$ = 0.94), recessive model (GG vs. GA+AA: OR = 0.81, 95% CI = 0.62–1.06, $P$ = 0.13 and codominant model (GG vs. AA: OR = 0.87, 95% CI = 0.53–1.42, $P$ = 0.58 and GA vs. AA: OR = 1.09, 95% CI = 0.73–1.65, $P$ = 0.67) (Fig f in S1 File, Table 2).

## Publication bias

The shape of the funnel plot did not suggest any obvious asymmetry between the seven SNPs and any myopia (see S2 File).

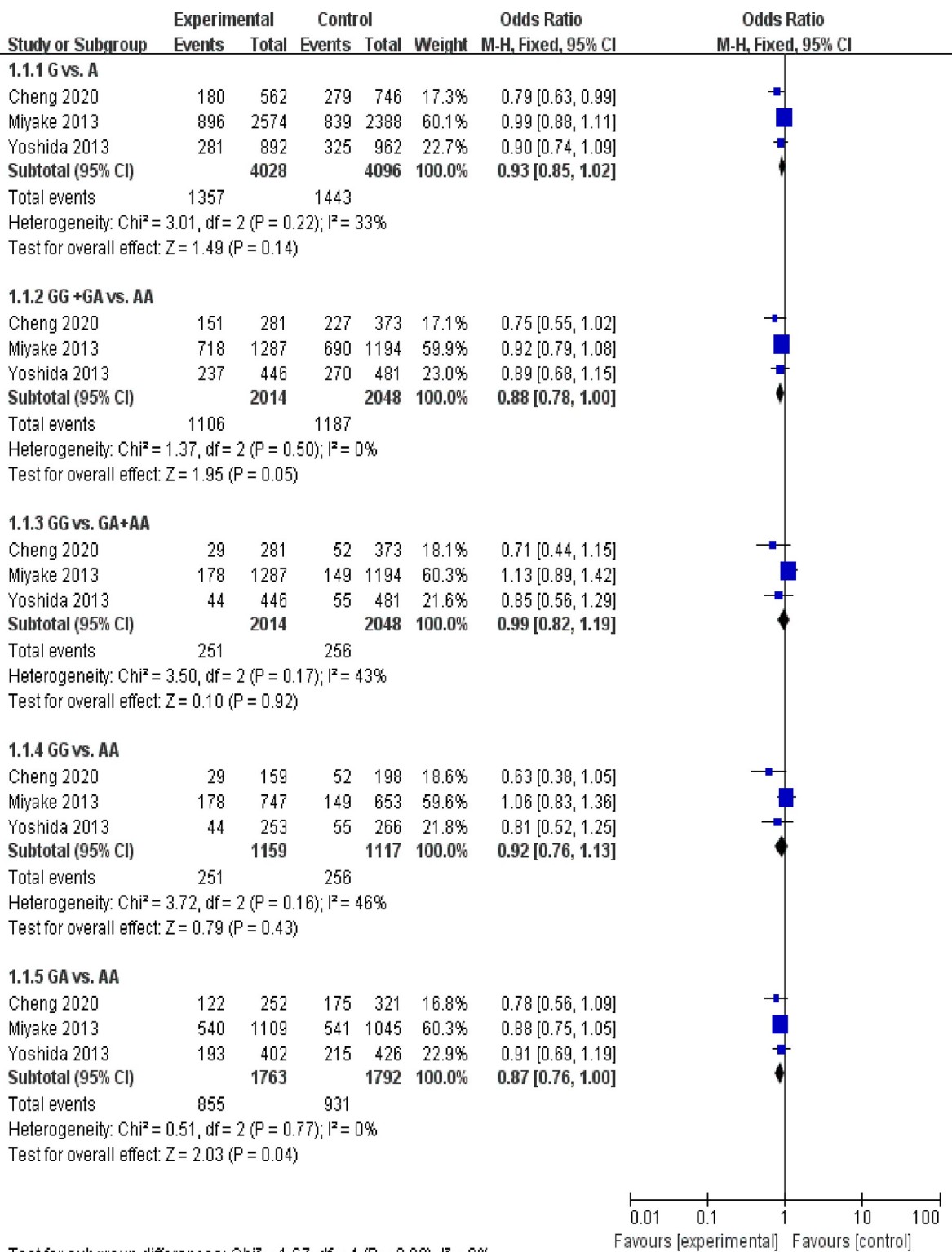

**Fig 2. Meta-analysis of the association of *IGF1* rs2162679 with any myopia.** Bars with squares in the middle represent 95% confidence intervals (95% CIs) and odds ratios (ORs). The central vertical solid line indicates ORs for the null hypothesis.

**Table 2. Main results of the pooled ORs between IGF1 SNPs and any myopia.**

| SNPs | Models Tested | | NO. study | Pooled OR | 95% CI | P | $P_Q$ | $I^2$ |
|---|---|---|---|---|---|---|---|---|
| **rs2162679** | Allelic model | G vs. A | 3 | 0.93 | 0.85–1.02 | 0.14 | 0.22 | 33% |
| | Dominant model | GG+GA vs. AA | 3 | 0.88 | 0.78–1.00 | 0.05 | 0.5 | 0% |
| | Recessive model | GG vs. GA+AA | 3 | 0.99 | 0.82–1.19 | 0.92 | 0.17 | 43% |
| | Codominant model | GG vs. AA | 3 | 0.92 | 0.76–1.13 | 0.43 | 0.16 | 46% |
| | | GA vs. AA | 3 | 0.87 | 0.76–1.00 | 0.04 | 0.77 | 0% |
| **rs6214** | Allelic model | A vs. G | 9 | 0.98 | 0.91–1.06 | 0.64 | 0.02 | 58% |
| | Dominant model | AA+AG vs. GG | 9 | 1.03 | 0.90–1.18 | 0.65 | 0.04 | 50% |
| | Recessive model | AA vs. AG+GG | 9 | 1 | 0.89–1.11 | 0.94 | 0.31 | 14% |
| | Codominant model | AA vs. GG | 9 | 1.02 | 0.87–1.20 | 0.82 | 0.11 | 39% |
| | | AG vs. GG | 9 | 1.02 | 0.90–1.15 | 0.73 | 0.17 | 31% |
| **rs12423791** | Allelic model | C vs. G | 6 | 0.95 | 0.81–1.11 | 0.51 | 0.005 | 70% |
| | Dominant model | CC+CG vs. GG | 6 | 0.96 | 0.80–1.16 | 0.68 | 0.03 | 61% |
| | Recessive model | CC vs. CG+GG | 6 | 0.92 | 0.73–1.15 | 0.45 | 0.14 | 40% |
| | Codominant model | CC vs. GG | 6 | 0.93 | 0.71–1.22 | 0.61 | 0.13 | 41% |
| | | CG vs. GG | 6 | 0.97 | 0.82–1.16 | 0.76 | 0.09 | 48% |
| **rs5742632** | Allelic model | C vs. G | 3 | 0.97 | 0.88–1.07 | 0.57 | 0.38 | 0% |
| | Dominant model | CC+CG vs. GG | 3 | 1.01 | 0.88–1.17 | 0.88 | 0.69 | 0% |
| | Recessive model | CC vs. CG+GG | 3 | 0.89 | 0.75–1.07 | 0.22 | 0.32 | 13% |
| | Codominant model | CC vs. GG | 3 | 0.91 | 0.75–1.12 | 0.38 | 0.39 | 0% |
| | | CG vs. GG | 3 | 1.04 | 0.90–1.21 | 0.59 | 0.86 | 0% |
| **rs10860862** | Allelic model | T vs. G | 3 | 1.02 | 0.91–1.14 | 0.8 | 0.7 | 0% |
| | Dominant model | TT+TG vs. GG | 3 | 1 | 0.87–1.16 | 0.98 | 0.76 | 0% |
| | Recessive model | TT vs. TG+GG | 3 | 1.06 | 0.84–1.35 | 0.62 | 0.89 | 0% |
| | Codominant model | TT vs. GG | 3 | 1.05 | 0.73–1.51 | 0.81 | 0.81 | 0% |
| | | TG vs. GG | 3 | 1 | 0.86–1.16 | 1 | 0.83 | 0% |
| **rs35766** | Allelic model | G vs. A | 3 | 0.93 | 0.74–1.16 | 0.51 | 0.01 | 78% |
| | Dominant model | GG+GA vs. AA | 3 | 0.95 | 0.69–1.31 | 0.77 | 0.02 | 74% |
| | Recessive model | GG vs. GA+AA | 3 | 0.81 | 0.65–1.00 | 0.05 | 0.24 | 29% |
| | Codominant model | GG vs. AA | 3 | 0.83 | 0.56–1.21 | 0.32 | 0.07 | 62% |
| | | GA vs. AA | 3 | 1.01 | 0.77–1.32 | 0.97 | 0.08 | 60% |
| **rs5742629** | Allelic model | G vs. A | 3 | 0.94 | 0.71–1.25 | 0.67 | 0.002 | 84% |
| | Dominant model | GG+GA vs. AA | 3 | 1.02 | 0.65–1.59 | 0.94 | 0.003 | 83% |
| | Recessive model | GG vs. GA+AA | 3 | 0.81 | 0.62–1.06 | 0.13 | 0.15 | 47% |
| | Codominant model | GG vs. AA | 3 | 0.87 | 0.53–1.42 | 0.58 | 0.02 | 75% |
| | | GA vs. AA | 3 | 1.09 | 0.73–1.65 | 0.67 | 0.01 | 78% |

## Discussion

As of August 4, 2021, the Online Mendelian Inheritance in Man (OMIM) database has listed 483 genetic factors associated with myopia. Additionally, two independent genome-wide association studies that involved large cohorts of refractive error patients identified loci at chromosome 15q14 and 15q25 [37, 38]. However, investigating the genetics of complex disorders such as any myopia remains a great challenge. Furthermore, the CREAM consortium conducted multi-center GWAS meta-analyses and identified susceptibility genes that affected diverse biological pathways [39], although they found no evidence of associations between *IGF1* SNPs and myopia. Extended axial length is known to be an important characteristic of the progress of myopia, which is associated with scleral remodeling. It is important to carefully analyze

genes in the scleral remodeling pathway. As mentioned above, *IGF1* could contribute to ocular enlargement by changing the structure of the sclera [23].

SNP rs2162679 of *IGF1* has been reported to be associated with several kinds of cancer [40–42], which reminds us that *IGF1* SNPs might play similar role in the onset or progesssion of myopia and cancer. In this study, our meta-analysis shows there is association between *IGF1* rs2162679 and any myopia in codominant model (GA vs. AA) and dominant model (GG+GA vs. AA). The genotype GA and GG+GA in rs2162679 have a lower risk of any myopia than those with the genotype AA. The G allele in this position may protect against the onset or progesssion of myopia.

Rs6214 is located within the intron of *IGF1*. In 2010, Metlapally et al. [43] and Zidan et al. [33] found that rs6214 was positively associated with any myopia/high-grade myopia after correcting for multiple testing. However, in other studies, no significant association for rs6214 was found using single marker analysis [27–32, 34, 35]. Zhuang et al. [31] and Zhao et al. [35] reported that rs12423791 was significantly associated with high myopia in a Chinese population. Although Mak et al. [29] found no association in a Chinese population, they identified a three-SNP haplotype consisting of rs12423791 with a significant association between high myopia and control participants using a variable-sized sliding-window strategy. The final results of this meta-analysis indicated that rs6214 and rs12423791 were not associated with any myopia. In this present study, we included three studies for meta-analysis of rs5742632, rs5742632, rs35766 and rs5742629 respectively. However, our analysis revealed no association between these SNPs and any myopia in genetic models.

Additionally, some other SNPs are notable, although we could not carry out meta-analysis. For example, rs12579077 and rs35767 were reported in the study of Mak et al. [29] in 2012, which are both located in the promoter region. Additionally, we have conducted SNP function prediction using the "SNPinfo Web Server", which suggests that the two SNPs may play important roles in susceptibility to high myopia. Additionally, rs12423791, rs7956547 and rs5742632 comprise a unit that may be associated with genetic susceptibility to high myopia in Chinese adults. Rs5742714 is located in the 3´-UTR of the *IGF1* gene. Variants in the 3´-UTR affect the binding region of microRNA, which plays an important role in disease by regulating translation of mRNA. Rs35766 is located in the 5´-near region. The 5´-near region may have a role in regulating the transcription of mRNA. In our present study, we found that rs35766 and rs1457601 were detected by one study [31] that suggested associations with high myopia. Although these two SNPs are located in the 5´-near region of the *IGF1* gene, which may play important roles in the process of transcriptional regulation, these associations need to be validated in further studies. Additionally, rs1457601 also is located in the 5´-near region. ALD map based on 1000 genome data provides potential evidence of a haplotypic effect between SNP rs1457601 and other SNPs, such as rs74633605, rs79196465 and rs79218426. Accordingly, the rs1457601 haplotypes also warrant future study.

There are several limitations to this present study. Firstly, the SNPs that we studied were all located in one chromosome according to existing data and haplotype analysis was not performed, which may have affected our results to some extent. It is necessary to pay more attention to haplotype analysis and SNPs on other chromosomes, especially those located in functional regions. Secondly, the major ethnic subjects was Asian, such as Japanese and Chinese. Besides, there are few studies on the polymorphism of any myopia, especially mild and moderate myopia. This two may affect the extrapolation of the conclusions. It is necessary to conduct further studies in other ethnic populations and subjects with different degrees of myopia. Thirdly, myopia is a complex disease affected by hereditary and environmental factors. Environmental factors may cause genetic changes. Gene-environment interactions should also be taken into consideration.

## Conclusion

In conclusion, this meta-analysis suggests that the G allele of the *IGF1* rs2162679 SNP is a potential protective factor for any myopia, which is worth further researches. Haplotype analysis and gene-environment interactions should also be taken into consideration.

## Supporting information

**S1 File. Meta-analysis of the association of other *IGF1* SNPs with any myopia.**
(DOCX)

**S2 File. Funnel plot analysis for publication bias.**
(DOCX)

**S3 File. Search strategy.**
(DOCX)

**S4 File. Prisma 2009 checklist.**
(DOCX)

## Acknowledgments

Our research team completed this meta-analysis independently and there are no people or groups to acknowledge to.

## Author Contributions

**Conceptualization:** Kang Wang, Yingxiang Huang, Yanling Wang.

**Formal analysis:** Bo Meng.

**Supervision:** Kang Wang, Yingxiang Huang, Yanling Wang.

**Writing – original draft:** Bo Meng.

**Writing – review & editing:** Bo Meng.

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
