## [Decision Letter · Decision Letter 0]

9 Jun 2022

PONE-D-22-06967The G allele of the IGF1 rs2162679 SNP is a potential protective factor for any myopiaPLOS ONE

Dear Dr. Meng,

Thank you for submitting your manuscript to PLOS ONE. After careful consideration, we feel that it has merit but does not fully meet PLOS ONE’s publication criteria as it currently stands. Therefore, we invite you to submit a revised version of the manuscript that addresses the points raised during the review process.

While both reviewers appreciate the contributions of this meta-analysis, but both have a number of comments that need to be addressed.  Although some are fairly minor, there are some inconsistencies with regard to the description of the outcomes in the Results and Discussion section and the need to correct for multiple testing, or at least discuss this and why it is not necessary, as well as some problems with the figures and tables.  Each of these need to be addressed.

We look forward to receiving your revised manuscript.

Kind regards,

James Fielding Hejtmancik, M.D., Ph.D.

Academic Editor

PLOS ONE

Journal Requirements:

2. Please confirm that you have included all items recommended in the PRISMA checklist including identifying the study as a meta-analysis or systematic review in the title.

No

NO authors have competing interests

5. Please ensure that you include a title page within your main document. You should list all authors and all affiliations as per our author instructions and clearly indicate the corresponding author.

6. Please amend your manuscript to include your abstract after the title page.

7. Please include a separate caption for each figure in your manuscript.

8. Please ensure that you refer to Figure 2 in your text as, if accepted, production will need this reference to link the reader to the figure.

9. Please include your tables as part of your main manuscript and remove the individual files. Please note that supplementary tables (should remain/ be uploaded) as separate "supporting information" files.

Additional Editor Comments:

While both reviewers appreciate the contributions of this meta-analysis, but both have a number of comments that need to be addressed. Although some are fairly minor, there are some inconsistencies with regard to the description of the outcomes in the Results and Discussion section and the need to correct for multiple testing, or at least discuss this and why it is not necessary, as well as some problems with the figures and tables. Each of these need to be addressed.

Reviewers' comments:

Reviewer's Responses to Questions

**Comments to the Author**

1. Is the manuscript technically sound, and do the data support the conclusions?

Reviewer #1: Yes

Reviewer #2: Yes

2. Has the statistical analysis been performed appropriately and rigorously? 

Reviewer #1: N/A

Reviewer #2: Yes

3. Have the authors made all data underlying the findings in their manuscript fully available?

Reviewer #1: Yes

Reviewer #2: Yes

4. Is the manuscript presented in an intelligible fashion and written in standard English?

Reviewer #1: Yes

Reviewer #2: Yes

5. Review Comments to the Author

Reviewer #1: In this manuscript the authors reported The G allele of the IGF1 rs2162679 SNP is a potential protective factor for any myopia. the project is well conceived, the following comments for consideration:

1. in this study the comparisons were done between the different groups and SNPs using analysis of variance, the authors need to do multiple testing to reject false hypotheses.

2. In the Results rs2162679, "significant association" should be changed to "suggestive association."

3. In the Results , the paragraph descriptions of rs2162679 that supporting file should be (Table 1, Table 2) instead of (Fig 1, Table 2), this paragraph not related with Fig 1.

4. in Abstract and Discussion the author said rs2162679

was significantly associated with any myopia in the codominant model (GA vs. AA: OR

= 0.87, 95% CI: 0.76-1.00) and the dominant model (GG+GA vs. AA: OR = 0.88, 95%

CI = 0.78-1.00).

but in results the author said rs2162679 "there were no significant associations for the allelic model (G vs. A: OR = 0.93,

95% CI: 0.85-1.02, P=0.14), dominant model (GG+GA vs. AA: OR = 0.88, 95% CI =

0.78-1.00, P=0.05), recessive model (GG vs. GA+AA: OR = 0.99, 95% CI = 0.82-1.19,

P=0.92 and codominant model (GG vs. AA: OR = 0.92, 95% CI = 0.76-1.13, P=0.43).", this is inconsistency, authors need to correct that.

5. the authors need add in figure legend for Fig 1 and Fig 2 .

Reviewer #2: Meng B, et al., presented a manuscript performing a meta-analysis to investigate the association of IGF1 SNPs and myopia. Authors analyzed nine studies and included 7 SNPs in this meta-analysis. They found that Rs2162679 was significantly associated with myopia in both the codominant and the dominant model, and the G allele of the IGF1 rs2162679 SNP is a potential protective factor for any myopia. This manuscript is well written and minor revision is needed to address the following points:

1. Please include “meta-analysis” in the title

2. Add a reference to Page 7 Line 12-13 for the “Myopia has already become the second most common cause of legal blindness”

3. Page 8 Line 9: delete “[22]”

4. Table 1 is truncated on the right side and is not fully shown in the manuscript

5. Check the format of all references.

6. PLOS authors have the option to publish the peer review history of their article (what does this mean?). If published, this will include your full peer review and any attached files.

Reviewer #1: No

Reviewer #2: No

---

## [Author Response · Author response to Decision Letter 0]

3 Jul 2022

The statement "The authors received no specific funding for this work." has been included in the cover letter.

The statement "The authors have declared that no competing interests exist." has been included in the cover letter.

---

## [Decision Letter · Decision Letter 1]

8 Jul 2022

The G allele of the IGF1 rs2162679 SNP is a potential protective factor for any myopia: Updated systematic review and meta-analysis

PONE-D-22-06967R1

Dear Dr. Meng,

We’re pleased to inform you that your manuscript has been judged scientifically suitable for publication and will be formally accepted for publication once it meets all outstanding technical requirements.

Kind regards,

James Fielding Hejtmancik, M.D., Ph.D.

Academic Editor

PLOS ONE

Additional Editor Comments (optional):

The authors have addressed each of the reviewers' comments.

Reviewers' comments:

Reviewer's Responses to Questions

**Comments to the Author**

1. If the authors have adequately addressed your comments raised in a previous round of review and you feel that this manuscript is now acceptable for publication, you may indicate that here to bypass the “Comments to the Author” section, enter your conflict of interest statement in the “Confidential to Editor” section, and submit your "Accept" recommendation.

Reviewer #1: All comments have been addressed

2. Is the manuscript technically sound, and do the data support the conclusions?

Reviewer #1: Yes

3. Has the statistical analysis been performed appropriately and rigorously? 

Reviewer #1: Yes

4. Have the authors made all data underlying the findings in their manuscript fully available?

Reviewer #1: Yes

5. Is the manuscript presented in an intelligible fashion and written in standard English?

Reviewer #1: Yes

6. Review Comments to the Author

Reviewer #1: the authors have been addressed all my concerns in the revised manuscript. overall, the results are complete and advance the field.

7. PLOS authors have the option to publish the peer review history of their article (what does this mean?). If published, this will include your full peer review and any attached files.

Reviewer #1: No

---

## [Editor Report · Acceptance letter]

12 Jul 2022

PONE-D-22-06967R1 

The G allele of the *IGF1* rs2162679 SNP is a potential protective factor for any myopia: Updated systematic review and meta-analysis 

Dear Dr. Meng:

I'm pleased to inform you that your manuscript has been deemed suitable for publication in PLOS ONE. Congratulations! Your manuscript is now with our production department. 

Kind regards, 

on behalf of

Dr. James Fielding Hejtmancik 

Academic Editor

PLOS ONE